# Assessment of the Level of Physical Activity and Mood in Students after a Year of Study in a Mixed Mode in the Conditions of Restrictions Resulting from the Pandemic

**DOI:** 10.3390/ijerph20054311

**Published:** 2023-02-28

**Authors:** Małgorzata Stefańska, Reninka De Koker, Jeroen Vos, Eveline De Wachter, Agnieszka Dębiec-Bąk, Agnieszka Ptak

**Affiliations:** 1Physiotherapy Faculty, Wroclaw University of Health and Sport Science, 51-612 Wrocław, Poland; 2Odisee Campus Brussel Terranova, 1000 Brussel, Belgium

**Keywords:** physical activity, depression, depressed mood, young adult, COVID-19

## Abstract

The COVID-19 pandemic has forced social isolation affecting all areas of life. It also affected the functioning of schools and universities. Many countries have introduced full or partial distance learning. The aim of the study was to assess the level of physical activity and student mood of the Faculty of Physiotherapy of the Academy of Physical Education in Wrocław (Poland) and students of the Faculty of Health of the ODISSE University in Brussels (Belgium) after a year of the study conducted in a mixed mode due to contact restrictions resulting from the COVID-19 pandemic and checking which of the analyzed factors increases the risk of depression to the greatest extent. Material and methods: 297 students from the 2nd to 4th year of full-time studies took part in the observation. The academic year 2020/2021 was assessed. Physical activity was assessed using the Global Physical Activity Questionnaire (GPAQ) recommended for this type of analysis by WHO. The GPAQ questionnaire enables the assessment of activity performed at work, movement, and leisure time and assesses the time of sitting or resting in a supine position. The Beck Depression Inventory was used to assess mental health. The subjects also completed a questionnaire concerning selected somatic features and describing their living conditions in the previous year. Results: In the group of Polish students, classes conducted in a completely remote mode accounted for about 50%, while in the group of Belgian students, about 75%. In the described period, 19% of students from Poland and 22% of students from Belgium were infected with COVID-19. The median of the results of the Beck Depression Scale in both groups was lower than 12 points (7 points in the AWF group and 8 points in the ODISSE group, respectively). A detailed analysis showed that in both study groups, more than 30% of students received results showing a depressed mood. A total of 19% of the surveyed students of the University of Physical Education and 27% of the ODISSE students were characterized by a result indicating mild depression. The results of the GPAQ questionnaire show that the total physical activity, including work/study, recreation, and mobility was 16.5 h a week for students from Poland and 7.4 h a week for students from Belgium. Conclusions: Both groups of subjects reached all the thresholds recommended by the WHO as a sufficient level of weekly physical activity. A group of students of the Faculty of Physiotherapy of the University of Physical Education in Wrocław was characterized by more than twice as high (statistically significant) level of weekly physical activity as compared to the group of participants from the ODISSE University in Brussels. In both study groups, more than 30% of students experienced a lowered mood of varying intensity. It is necessary to monitor the mental state of students and, in the event of obtaining control results at a similar level, to implement psychological assistance for willing participants.

## 1. Introduction

In the spring of 2020, mass cases of the SARS-CoV-2 virus causing a highly infectious respiratory infection COVID-19 were observed all over the world. Due to the rapid spread of the virus, in order to reduce the transmission of infections, the World Health Organization (WHO) declared a pandemic and issued recommendations for the governments of individual countries on how to identify and isolate people with suspected infections [1]. Recommendations included mass testing, quarantine, and limiting human contact. In many countries, it was ordered to limit the functioning of workplaces, hotels, and gastronomy. A ban on organizing mass events and special rules regarding the use of public spaces were also introduced [2]. Borders were closed, which significantly limited movement between individual countries. Scientific research has shown the impact of restrictions related to the pandemic on the social life of people of all ages. There is documented social isolation, loneliness, and lower quality of life of elderly individuals during the COVID-19 pandemic [1,2,3]. Changes were observed in work practices, lifestyle, physical activity, and well-being among desk workers during shelter-at-home restrictions [4,5,6]. A negative impact has been proven of COVID-19 confinement on the lifestyle behaviors of children and adolescents [7,8,9] and also on the quality and quantity of sleep time. [10]. The negative impact of the pandemic on mental and physical health has also been demonstrated [11,12,13,14,15,16].

The COVID-19 pandemic has forced social isolation affecting all areas of life. It also affected the functioning of schools and universities. Many countries have introduced full or partial distance learning. These actions significantly limited the spread of the virus but intensified the feeling of loneliness [17]. Limiting interpersonal contact harmed well-being and physical and mental health [18,19]. Yao Zhang et al. (2022), conducting observations among Chinese students, showed that quarantine and prolonged isolation caused negative emotions manifested, among others, by an increased perceived level of anxiety and stress [20].

Isolation has significantly limited opportunities to participate in various forms of physical activity. Wunsch et al., 2022 in a meta-analysis of 57 different studies conducted in 14 countries covering the period of the COVID-19 pandemic, showed a significant decrease in activity levels in 32 studies, while only five showed a significant increase. Fourteen studies produced mixed results [21].

The advantages of physical activity are widely described in the literature on the subject. Physical activity provides physical health benefits, for example, improving functional performance, reducing the risk of somatic and mental diseases, improvement of body composition, and weight loss [22,23].

Mental well-being has a huge impact on all spheres of life. Many earlier studies showed the link between mental health with physical activity, sleep quality, and quality of life [24,25]. A beneficial impact on physical and mental health was also confirmed in people participating in exercises conducted indoors [26,27]. It has also been proven that physical activity, by improving the quality of sleep, helps significantly reduce the risk of mental illness [28].

Until now, little research included the assessment and possibilities of improving people’s mental health during the COVID-19 pandemic. Observations conducted most often referred to the Chinese population and were concerned with the assessment of mental health and the mental adaptation of society [29] and the occurrence of symptoms related to stress, depression, and a sense of anxiety [18,19,30,31,32,33] of the impact of demographic and sex factors and the age of the examined on the frequency of occurrence of mental disorders [19,33].

It was assumed that restrictions related to social restrictions would have a significant impact on the level of mood and the level of activity of respondents. Due to this, the study aimed to assess the level of physical activity and mood of students of the Faculty of Physiotherapy of the Academy of Physical Education in Wrocław (Poland) and students of the Faculty of Health of the ODISSE University in Brussels (Belgium) after a year of mixed-mode study due to contact restrictions resulting from the COVID-19 pandemic and checking which of the analyzed factors increases the risk of depression to the greatest extent.

## 2. Materials and Methods

### 2.1. Study Group

A total of 297 students from the 2nd to the 4th year of full-time studies took part in the observation (219 students of the Faculty of Physiotherapy of the University of Physical Education in Wrocław and 78 students of the Faculty of Health of the ODISSE University in Brussels). Recruitment for the research consisted in providing students with a link to online questionnaires via student social networking sites. Participation in the project was voluntary. Students were informed about the purpose of the research and had access to their results. The inclusion criterion for the project included participation in full-time teaching and a declared lack of chronic physical and mental illnesses in the year preceding the study. The study groups did not differ statistically significantly in terms of age, year of study, and basic anthropometric parameters. A detailed description of the study groups is provided in Table 1.

The study was conducted In the form of a semi-structured interview with the questionnaire, without any interventional and experimental structure, with the consent of the participants, and under the ethical and legal supervision of the Department of Physiotherapy at the University of Physical Education in Wrocław. The study was conducted following the Declaration of Helsinki.

### 2.2. Research Methods

Physical activity was assessed using the Global Physical Activity Questionnaire (GPAQ) recommended for this type of analysis by WHO in Polish, English, or French [34,35,36,37]. The Beck Depression Inventory (BDI) [38,39,40] was used to assess mental state. The academic year 2020/2021 was assessed. The study was conducted in June 2021. The subjects also completed a questionnaire concerning selected somatic features and describing their living conditions in the previous year.

The GPAQ questionnaire used to assess the level of physical activity is a synthetic combination of the short and long versions of the IPAQ questionnaire. It is limited to 16 questions that enable independent assessment of activities performed during work/study (questions 1–6), mobility (questions 7–9), and free time (questions 10–15), as well as assessing sitting or resting time in lying down (question 16). Respondents, answering individual questions, determined the number of active days in a typical week and the number of active minutes in a typical day and also selected the intensity level. High-intensity activities were defined as activities that required a lot of physical effort, causing a significant increase in breathing and heart rate. Moderate-intensity activities were defined as requiring moderate physical exertion, causing a slight increase in breathing and heart rate. A study of the reliability of the Global Physical Activity Questionnaire (GPAQ) in 9 countries showed moderate to significant reliability (Kappa 0.67 to 0.73; Spearman’s rho 0.67 to 0.81) [41]. The verification of the Polish version of GPAQ has not been published yet.

The Beck Depression Inventory is a self-assessment screening tool. It consists of 21 questions describing the attitude towards oneself and other people, satisfaction with life and events that have taken place recently and the emotions that accompanied them, as well as involvement in undertaken activities, feeling pleasure and joy from achieved successes and positive situations. Each question has 4 possible answers scored from 0 to 3 points. The criteria for evaluating the results include summing up all points scored. A score of 0 to 11 indicates no depression, 12 to 19 indicates mild depression, 20 to 25 indicates moderate depression, and more than 26 indicates severe depression. The BDI is characterized by a very high internal consistency; Cronbach’s alpha for the entire standardization sample is 0.93, and for patients with depression, 0.92 [42].

### 2.3. Statistical Analysis

The distribution of quantitative variables was verified by The Shapiro--Wilk test. Due to the non-normal distribution of quantitative variables and the ordinal or nominal nature of the remaining variables, the median was used as a measure of central tendency and the interquartile range as a measure of dispersion. The significance of differences between the study groups was verified using the Mann--Whitney U test or the Chi-square test. Logistic regression was performed to indicate which factors had the greatest impact on mental well-being assessed by the BDI. Additionally, to determine the quantity of the effect of differences between examined groups, a corrected Cohen’s d test was used for continuous and ordinal variables. The effect size of interaction was calculated by Eta squared (η2) and then transformed to Cohen’s d value [43]. The values of Cohen’s d test ≥ 0.8 indicated great strength of the observed effect, ≥ 0.5 moderate effect, ≥ 0.2 weak effect, and <0.02 no effect [44]. Cramer’s V coefficient was used to calculate the effect size of the χ2 test with more than one degree of freedom (categorical variables). Cramer’s V value is in the range of 0–1. The closer it is to 0, the weaker the strength of the relationship between the examined features, and the closer it is to +1, the stronger the strength of the studied relationship is [45]. In a detailed interpretation, Cramer’s V ≥ 0.5 indicates a significant connection effect, ≥0.03 is moderate, ≥0.1 1 small effect, and <0.1 no effect. Calculations were made in Statistica 13.3 and PQ Stat 1.8.2 and statistical calculators http://www.psychometrica.de/effect_size (access date 15 February 2023).

The significance level was *p* < 0.05.

## 3. Results

The analysis of the obtained results showed that in both study groups, most of the academic classes took place in a completely remote or hybrid mode in the period under consideration. Hybrid teaching was defined as a situation in which part of the course was conducted face-to-face and part remotely. The observed difference between the study groups showed statistical significance (Table 2).

Observations showed that in the described period, about 20% of students from Poland and Belga were under COVID-19 infection, and the statistical significance between the groups resulted from the different amounts of lack of answers to this question. Research groups differed significantly in residence. Most students from Poland lived alone (in the dorm, student apartment, or their apartment), while most Belgian students lived in family homes. No significant changes in the place of residence were observed in both study groups (Table 3).

The median of the results of the Beck Depression Inventory in both groups was lower than 12 points (respectively 7 ± 10 points in the AWF group and 8 ± 11 points in the ODISSE group, *p* = 0.0745). However, a detailed analysis showed that in both study groups, more than 30% of students received results showing a depressed mood. In total, 19% of the surveyed students of the University of Physical Education and 27% of the ODISSE students were characterized by a result indicating mild depression. Respectively, 6% and 9% of students received a result indicating the possibility of moderate depression, and 6% and 9% of students had a result indicating severe depression (Table 4 and Table 5).

Analysis of students’ physical activity was divided into four fields. Statistically significant differences between groups regarding the level of physical activity related to science/work and recreation were observed, as well as the time of sitting/lying position. Students from Poland declared higher physical activity, both daytime and weekly, related to learning and work as well as recreation, and more than twice shorter sitting/lying down compared to students from Belgium. However, there were no significant differences between groups in the level of activity related to the movement and length of sleep (Table 6, Table 7, Table 8 and Table 9).

The summary analysis covering physical activity related to work, recreation, and travel showed more than twice the values indicated by the students of the University of Physical Education in comparison to the students of ODISSE, both on a daily and weekly basis (Table 10).

In order to determine the strength of the observed phenomena, the effect size was calculated for each comparison between the study group and the control group. Depending on the analyzed variable and the type of comparison, Cohen’s d or Cramer’s V was used. In most analyses showing the statistical significance of differences between the groups, the effect size value indicated a medium or high effect (0.558 ≤ Cohen’s d ≥ 1.162). The exception was the effect size values calculated for the variables for which the Chi-square test was used to compare them. Despite the statistical significance of the Chi-square test, the calculated values of Cramer’s V indicated a weak or medium effect (0.171 ≤ Cramer’s V ≥ 0.370) resulting from high intra-group variability and different sizes of the compared study groups. In analyses where statistical significance was not confirmed, a small size effect was demonstrated regardless of the test used (Table 1, Table 2, Table 3, Table 4, Table 5, Table 6, Table 7, Table 8, Table 9 and Table 10).

The analysis of logistic regression showed that only the length of sleep has a significant impact on the occurrence of the risk of depression. The quotient of opportunities calculated for this model indicates that reducing sleep length by 1 h a week increases the risk of depression 0.92 times. The quotient of opportunities is calculated for individual values. It should be noted that after their multiplication, the chance of changing the explanatory variable is also multiplied, i.e., shortening the length of sleep by 10 h a week will increase the risk of depression by 9.2 times. The statistical significance of the model was also confirmed in the analysis of the influence of the subjects. The female sex has been shown to increase the chance of depressive symptoms 0.36 times (Table 11).

## 4. Discussion

The infection of the immune system can cause psychopathologies, which was confirmed in previous observations of the coronavirus epidemic [33,46,47]. Severe acute respiratory failure (COVID-19) caused by coronavirus may be associated with psychiatric implications [48]. Coronavirus can cause psychopathological consequences through direct viral infection of the central nervous system (OUN) or indirectly through an immune response [49]. Trials made on postmortem, animals, in vitro, and cell farms have shown that coronavirus is potentially neurotropic and can cause neuron damage [50].

What is more, previous studies on psychological reactions among the general Chinese population [51] or university students [52] during the epidemic of severe respiratory failure (SARS) showed that people who were quarantined or indirectly exposed to SARS had obtained insufficient social support and applied avoidance strategies, more often experienced psychological symptoms [53]. During the COVID-19 pandemic, there was an increasing percentage of all forms of mood disorders: depression by 20%, anxiety by 35%, and stress by 53% in a combined research population of 113,285 people in 16 studies conducted in China, India, Spain, and Italy [49]. In Canada, where an online survey was conducted among 1803 participants, the percentage of participants who indicated that their anxiety was high to extremely high quadrupled (from 5% to 20%), and the number of participants with high self-reported depression more than doubled (from 4% to 10%) since the onset of COVID-19. In the USA, where similar research was taken, depression syndromes were more than 3-fold higher during COVID-19 compared with before COVID-19 [50]. In Ireland during the pandemic time, generalized anxiety disorders GAD (20.0%), depression (22.8%), and generalized anxiety disorders or depression (27.7%) were higher [54].

Farabaugh et al. (2012) conducted extensive research on the mental health of students in the period before the COVID-19 pandemic. Using the Beck Depression Inventory (BDI), they showed depressive symptoms in 13% (BDI ≥ 13) of 898 students at 3 US universities with an average age of 20.07 years (±1.85) [51]. In this research, the criteria proposed by Beck (1961) [38] were used, and for the threshold value for the presence or absence of symptoms of depression, 12 points was assumed. To compare the results with the values presented by Farabaugh (2012) [51], a re-analysis was performed, and BDI values greater than or equal to 13 points were obtained in 26.5% of the surveyed students from Poland and 38.5% of students from Belgium. The variability of the results of screening tests for depression was demonstrated by Rotenstein et al. (2016) in a systematic review of 184 publications from 43 countries. In studies involving students of medical schools, depressive symptoms were found in 27.2% of the respondents, with the range of the obtained results ranging from 9.3% to 55.9% [52]. Jaworska et al. 2014 studied the frequency of depressive symptoms in 153 students of the University of Physical Education in Wrocław (one of the universities where their research was conducted, but involving a different group of participants). In the period before the pandemic, they observed symptoms of depressed mood (BDI ≥ 12) in 29% of respondents, including severe symptoms of depression in 8% (BDI > 20) [55]. In this research, depressed mood (BDI ≥ 12 [38]) was observed in 31% of students from Poland and 45% from Belgium. The observed between-group difference showed statistical significance, but due to high intra-group variability, different group sizes, and overall small size, the determined strength of this comparison showed a weak effect. The above examples of comparisons of screening results covering depressive symptoms in students during the COVID-19 pandemic period and in the preceding period did not show differences at the level of the results obtained using the BDI questionnaire due to the large diversity.

Rotenstein et al. (2016) indicate that only 15.7% of students whose screening for depression gave a positive result reported treatment [52]. These discoveries are disturbing, considering that the development of depression is associated with a short-term risk of suicide, as well as a higher long-term risk of future depressive and incidence episodes [56].

The latest WHO recommendations regarding physical activity indicate that adults should perform at least 150–300 min of aerobic physical activity with moderate intensity or at least 75–150 min of intensive aerobic physical activity in a week or an equivalent combination of moderate and high-intensity exercises. According to WHO, adults should also do exercises to strengthen muscles with moderate or greater intensity that involve all main muscle groups for two or more days a week because they provide additional health benefits. In addition, the time spent in a sitting position should be limited. The replacement of a sitting lifestyle with physical activity of any intensity (including low intensity) provides health benefits [57]. During the last two years, there have been many scientific reports indicating a significant limitation of physical activity caused by restrictions resulting from insulation. Ming-Qiang Xiang et al., in their research conducted on a population of 1421, discovered that during the COVID-19 outbreak, about 52.3% of Chinese college students had inadequate physical activity. The rates of anxiety and depression symptoms were 31.0 and 41.8%, respectively [58]. In addition, Castañeda-Babarro et al. (2020) among the adult population of Spain (3400 people), especially adolescents, students, and very active people, observed a decrease in the daily, self-reported activity level during the COVID-19 pandemic [59]. Luciano et al. (2021), based on two studies of 714 medical students using the International Physical Activity Questionnaire Short Form (IPAQ), proved a decrease in physical activity, increased sitting time, and shorter sleep compared to the pre-pandemic period [60].

In this research, the level of activity of students was assessed once, after a year of study, in remote and mixed mode using the GPAQ questionnaire. Both Polish and Belgian university subjects achieved all levels of activity recommended by the WHO [57], and the sum of activities related to work, recreation, and transport was 91MET-hours/week (5460MET-min/week) and 38MET-hours/week, respectively (2280MET-min/week). In comparison with the results reported by other researchers, the values obtained in our own research indicate the high physical activity of the study group. The equivalence of the GPAQ and IPAQ questionnaires is described as moderate to strong (range 0.45 to 0.65) [41] and even very strong (r = 0.79–0.94, *p* < 0.001) [61], which allows for the comparison of some results obtained in both questionnaires. The final IPAQ SF score (MET-min/week) can be compared with the sum of work, recreation, and transportation activities of the GPAQ questionnaire (MET-min/week).

Luciano et al., 2021 assessed the physical activity of medical students using IPAQ SF. During the Before Lockdown period, they obtained students of 6 years, 1588 MET-min/Week, and 1200 MET-min/Week in students 1–5 years and 1170 met-min/week and 960 met-min/week, respectively, during the lockdown period [60]. Wattanapisit et al. (2016) obtained in single research in the study group of 285 medical students in Southern Thailand the level of activity estimated by the GPAQ in the range of 0 -5640 MET -min/Week [62]. Jalal et al. (2021) determined the level of activity of 628 students in Saudi Arabia before and during restrictions related to COVID-19, obtaining GPAQ scores equal to 1123 met-min/week and 10,821 met-min/week [63]. The quoted examples confirm the high level of physical activity observed in this research. However, the nature of the conducted research presents the results obtained during the period of restrictions in the period of study in stationary mode. The study groups statistically differed significantly in a sitting position. On average, a 3-h sitting/lying time was observed in a group of students from Poland and 7 h in students from Belgium. The significance of differences was confirmed by high effect size. The difference may be due to participation in various forms of didactic classes. In students from Poland at the observed time, 50% of classes took place in a hybrid form, which meant that about half of these classes took place in the form of practical classes in direct contact. In the research of other authors, the median number of hours spent in the sitting position was in the range of 7–10 h [60,63,64,65].

According to a metanalysis conducted by Rebar et al. based on a total of 92 studies with 4310 participants researching physical activity-depression and 306 studies researching 10,755 participants, physical activity–anxiety correlation showed that physical activity reduced depression by a medium effect and anxiety by a small effect [66]. In research conducted in Ireland, citizens aged 18–34 had significantly lower levels of COVID-19-related anxiety than adults aged 65 or older. What is important is, screening positive for GAD (Generalized Anxiety Disorder Scale) or depression was associated with younger age, female sex, loss of income due to COVID-19, COVID-19 infection, and higher perceived risk of COVID-19 infection [52].

In the studies of Merellano-Navarro (2022), the quality of sleep and physical activity in students of physical education pedagogy during the COVID-19 pandemic were analyzed. Good sleep quality is associated with a high level of physical activity. The prevalence of good-quality sleep was low and significantly lower in women. Male gender and a high level of physical activity during quarantine have a positive effect on good sleep quality, regardless of age. Maintaining healthy habits such as sleep and physical activity while studying is important due to the short- and long-term health repercussions; furthermore, there is a gender difference in both aspects that need to be taken into account in this population [67]. Jaworska et al. (2014) also proved that the average mood level of male and female students who regularly engaged in physical activity at least twice a week was significantly better than those who were less physically active. In addition, the authors observed that depressed mood affected women more often than men [55]. In the study, an attempt was made to assess the variable most conducive to depressive symptoms. Using logistic regression, it was proved, as in previous reports, that the probability of depressive symptoms is higher in the group of female. There was also a significant link between the length of sleep and the likelihood of depression. However, the created regression model did not confirm the significant impact of physical activity, type of classes, history of COVID-19 infection, or place of residence.

## 5. Limitations

The basic limitations of this study are the relatively small sample size and the observations carried out only in two academic centers. Collecting more data from different countries will enable a better understanding of global trends related to the impact of social restrictions on the psychophysical behavior of students. The size of the groups makes it impossible to analyze the impact of more factors on the level of mood. The analyses did not take into account, for example, socio-economic factors.

## 6. Conclusions

The primary analysis of the central tendency and range of all Beck Inventory scores showed no values indicative of a depressed mood. The median of the total BDI score in both study groups did not exceed 12 points. However, detailed analysis in both study groups showed that more than 30% of students experienced a lowered mood of varying intensity. It is necessary to monitor the mental state of students and, in the event of obtaining control results at a similar level, to implement psychological assistance for willing participants.

All study participants showed no reduction in weekly physical activity regarding the normative values. Both groups of subjects reached all the thresholds recommended by the WHO as a sufficient level of weekly physical activity. The group of students from Poland was more than twice as high (statistically significant) in the level of weekly physical activity as compared to the group of participants from Belgium.

The conducted analysis showed that the gender of the respondents and the number of hours of sleep per week had a significant impact on the appearance of depressive symptoms. However, no significant relationship was found between the result showing a depressed mood and the group of respondents, age, year of study, history of COVID-19 infection, level of physical activity, or place of residence. The lack of a significant relationship between mood and other variables may result from the small number of respondents and the high internal variability of the results.

## Figures and Tables

**Table 1 ijerph-20-04311-t001:** Research Group.

	AWF Wroclaw (Poland)		ODISSE Brussels (Belgium)			
N (Women/Men)	219 (143/76)		78 (66/12)			
	Median	IQR	Median	IQR	*p* U MW Test	Cohen’s d
age [years]	20.00	2.00	21.00	2.00	0.1134	0.234
year of study	2.00	1.00	2.50	1.00	0.1567	0.220
body height [cm]	170.00	16.00	168.00	9.00	0.0660	0.478
body weight [kg]	63.00	18.00	63.00	14.00	0.4514	0.393
BMI	22.04	3.17	21.60	4.01	0.7108	0.359

*p* < 0.05.

**Table 2 ijerph-20-04311-t002:** Method of carrying out classes in the academic year 2020/2021.

	AWF Wroclaw (Poland)		ODISSE Brussels (Belgium)		Ch^2^ Test/ p Chi^2^	Cramer’s V
	N	%	N	%		
Remote	110	50.2	60	76.9	40.70/ <0.0001 *	0.370
Hybrid	109	49.8	12	15.4		
In touch	0	0	6	7.7		
summary	Mediana	IQR	Mediana	IQR	P U MW	Cohen’s d
Remote classes %	70.00	15.00	80.00	20.00	<0.0001 *	0.716
Contact classes %	30.00	15.00	20.00	20.00	<0.0001 *	0.716

* *p* < 0.05.

**Table 3 ijerph-20-04311-t003:** Characteristics of the study groups.

		AWF Wroclaw (Poland)		ODISSE Brussels (Belgium)		Ch^2^ Test/ p Chi^2^	Cramer’s V /Cohen;s d
		N	%	N	%		
Infection COVID-19	Yes	42	19.2	17	21.8	8.64/ 0.0133 *	0.171
	No	131	59.8	56	71.8		
	No answer	46	21	5	6.4		
Place of residence	With parents	65	29.7	57	73.1	44.75/ <0.0001 *	0.8424
	Alone	154	70.3	21	26.9		
Change of residence	No	185	84.0	68	87.2	2.69/ 0.2603	0.095
	Yes. I started living with my parents	26	12.0	5	6.4		
	Yes. I started living alone	8	4.0	5	6.4		

* *p* < 0.05.

**Table 4 ijerph-20-04311-t004:** Beck Depression Inventory (BDI) results-Lowering the mood yes/no.

	No Symptoms of Depression (BDI < 12)		Depressive Symptoms (BDI ≥ 12)		Ch^2^ Test/ *p* Chi^2^	Cohen’s d
	N	%	N	%		
AWF we Wrocławiu (Poland)	151	68.9	68	31.1	5.18 0.0276 *	0.267
ODISSE w Brussels (Belgium)	43	55.1	35	44.9		

* *p* < 0.05.

**Table 5 ijerph-20-04311-t005:** Detailed results of the Beck Depression Inventory-summary.

	AWF Wroclaw (Poland)		ODISSE Brussels (Belgium)		Ch^2^ Test/ *p* Chi^2^	Cramer’s V
	N	%	N	%		
no depression (BDI < 12)	151	68.9%	43	55.1%	4.94 0.1756	0.129
mild depression (12–19)	42	19.2%	21	26.9%		
moderate depression (20–25)	12	5.5%	7	9.0%		
severe depression (26–63)	14	6.4%	7	9.0%		

BDI-Beck Depression Inventory; *p* < 0.05.

**Table 6 ijerph-20-04311-t006:** Physical activity related to work/study (GPAQ).

WORK/STUDY Heavy + Moderate	AWF Wroclaw (Poland)		ODISSE Brussels (Belgium)		*p* U MW Test	Cohen’s d
	Median	IQR	Median	IQR		
Number of hours a day	0.50	4.50	0.00	0.00	<0.0001 *	0.704
Number of hours a week	0.50	14.00	0.00	0.00	<0.0001 *	0.715
Met per day	2.00	24.00	0.00	0.00	0.0001 *	0.689
Met per week	4.00	72.00	0.00	0.00	<0.0001 *	0.700

MET–energy equivalent 1 MET = 1 kcal × kg^−1^ × h^−1^, * *p* < 0.05.

**Table 7 ijerph-20-04311-t007:** Physical activity related to traveling from place to place (GPAQ).

TRAVELLING	AWF Wroclaw (Poland)		ODISSE Brussels (Belgium)		*p* U MW Test	Cohen’s d
	Median	IQR	Median	IQR		
Number of hours a day	0.50	1.00	0.50	0.75	0.2774	0.409
Number of hours a week	3.00	7.00	2.45	3.00	0.2760	0.409
Met per day	2.00	4.00	2.00	3.00	0.2774	0.409
Met per week	12.00	28.00	9.80	12.00	0.2760	0.409

MET–energy equivalent 1 MET = 1 kcal × kg^−1^ × h^−1^, *p* < 0.05.

**Table 8 ijerph-20-04311-t008:** Physical activity related to recreation activity (GPAQ).

RECREATION Heavy + Moderate	AWF Wroclaw (Poland)		ODISSE Brussels (Belgium)		*p* U MW Test	Cohen’s d
	Median	IQR	Median	IQR		
number of hours per day	2.00	2.00	1.10	2.00	0.0071 *	0.558
number of hours per week	5.00	8.40	2.75	6.00	0.0034 *	0.581
MET per day	12.00	16.00	8.00	16.00	0.0058 *	0.565
MET weekly	32.00	56.00	16.00	46.00	0.0027 *	0.588

MET energy equivalent 1 MET = 1 kcal × kg^−1^ × h^−1^, * *p* < 0.05.

**Table 9 ijerph-20-04311-t009:** Daily and weekly sleep time and time spent sitting and lying down (GPAQ).

Activity	AWF Wroclaw (Poland)		ODISSE Brussels (Belgium)		*p* U MW Test	Cohen’s d
	Median	IQR	Median	IQR		
Sitting/lying-number of hours per day	3.00	3.00	7.00	3.00	0.0001 *	1.162
Sitting/lying-number of hours per week	21.00	21.00	49.00	21.00	0.0001 *	1.162
Amount of sleep per day	7.00	1.00	7.50	1.00	0.0664	0.478
Amount of sleep per week	49.00	7.00	52.50	7.00	0.0664	0.478

* *p* < 0.05.

**Table 10 ijerph-20-04311-t010:** Total score including physical activity related to work/study, traveling, and recreation (GPAQ).

SUM WORK + TRANSPORT + RECREATION	AWF Wroclaw (Poland)		ODISSE Brussels (Belgium)		*p* U MW Test	Cohen’s d
	Median	IQR	Median	IQR		
Number of hours a day	4.75	7.00	2.33	3.00	<0.0001 *	0.770
Number of hours a week	16.50	25.00	7.35	9.00	<0.0001 *	0.807
Met per day	24.00	32.00	14.00	16.40	<0.0001 *	0.734
Met per week	91.20	134.40	38.00	56.80	<0.0001 *	0.795

MET–energy equivalent 1 MET = 1 kcal × kg^−1^ × h^−1^, * *p* < 0.05.

**Table 11 ijerph-20-04311-t011:** Logistic regression calculated for the risk of depression (BDI ≥ 12).

Variable–BDI 0–BDI < 12 (No Depression) 1–BDI ≥ 12 (Risik of Depression)	coef. B	*p*-Value	Odds Ratio	±95% CI
Group	0.25	0.5314	1.28	0.59–2.76
Sex	−1.01	0.0048 *	0.36	0.18–0.73
Year of study	−0.19	0.3160	0.82	0.56–1.2
Classes in remote %	−0.01	0.4654	0.99	0.97–1.01
COVID-19 infection	−0.09	0.7954	0.91	0.47–1.79
Place inhabited	0.32	0.3494	1.37	0.71–2.66
Total activity	0.01	0.2380	1.01	0.99–1.02
Sitting/lying for a week	0.01	0.0962	1.01	1–1.03
Sleep a week	−0.08	0.0004 *	0.92	0.88–0.97
Logistic regression coef. *p*	0.0003 *			

BDI-Beck Depression Inventory; Group: AWF = 0, ODISSE = 1; Gender: women = 0, men = 1; COVID-19 infection: no = 0, yes = 1; Place inhabited: independent = 0, in the family home = 1; Total activity-sum: work + travelling + recreation; * *p* < 0.05.

## Data Availability

The data is stored in the archive of the Research Laboratory of the Faculty of Physiotherapy, Wroclaw University of Health and Sport Science.

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
