# Peer review of "Assessment of the Level of Physical Activity and Mood in Students after a Year of Study in a Mixed Mode in the Conditions of Restrictions Resulting from the Pandemic"

_ijerph, 2023, doi:10.3390/ijerph20054311_

Round 1

Reviewer 1 Report

Dear Authors.

I express my sincere thanks for the opportunity to review your paper on students' physical activity levels and mental well-being.

I already had indications but I am surprised to confirm in the paper how the COVID-19 pandemic has affected students and how they have adapted to the restrictions on physical contact. I find the paper provides valuable information on the impact of the pandemic on students and their state of mind. The research also shows the need to monitor the mental state of students and provide psychological assistance in case of similar outcomes.

However, I consider that the manuscript can be improved in some aspects even though I consider the manuscript viable for future publication. 

It is suggested that the Introduction section briefly introduce the context of the Covid-19 pandemic and its effects on social and school/university life. Although this would not require a great deal of length, it would help to provide evidence for future epidemics and similar situations. Although the publisher's parameters may not allow for a Literature Review section, it may be advisable to expand on what is discussed in the Introduction section.

The Methods and Discussion have a consistent structure. I suggest adding data on future research. Due to the limited local context of the questionnaire, it may be necessary to identify in the discussion the need for more data should be collected from different countries in order to gain a better understanding of global trends related to this topic. 

The Conclusions may be somewhat sparse. They should be expanded briefly. In addition, I suggest that paragraphs should be at least 5-6 lines long. In the Conclusions, there are paragraphs with only two lines. 

Finally, regarding the Title of the manuscript, I consider it is too long, and an effort should be made to reduce its length to 12-16 words. This will improve its positioning. Also, I ask for a brief discussion among the manuscript’s authors on the convenience of keeping the word Covid-19 in the title. As I have commented in the Introduction section, please join the comments made in the Introduction section. After all, Covid-19 may no longer be a topic of academic interest, but similar situations may arise in the future. 

Finally, please consider whether the citation form of the references is correct. You are using brackets instead of square brackets. Check this term with the journal template. This would be the main reason for considering the manuscript as "Major revisions".

Good luck with the revisions!

Author Response

Thank you for your time and attention, we found all the suggestions valuable and we have done our best to improve our manuscript. Below, you will find our answer and amendments changes.

I express my sincere thanks for the opportunity to review your paper on students' physical activity levels and mental well-being.

I already had indications but I am surprised to confirm in the paper how the COVID-19 pandemic has affected students and how they have adapted to the restrictions on physical contact. I find the paper provides valuable information on the impact of the pandemic on students and their state of mind. The research also shows the need to monitor the mental state of students and provide psychological assistance in case of similar outcomes.

However, I consider that the manuscript can be improved in some aspects even though I consider the manuscript viable for future publication.

It is suggested that the Introduction section briefly introduce the context of the Covid-19 pandemic and its effects on social and school/university life. Although this would not require a great deal of length, it would help to provide evidence for future epidemics and similar situations. Although the publisher's parameters may not allow for a Literature Review section, it may be advisable to expand on what is discussed in the Introduction section. – according to the suggestion we added to lines 47-64

The Methods and Discussion have a consistent structure. I suggest adding data on future research. Due to the limited local context of the questionnaire, it may be necessary to identify in the discussion the need for more data should be collected from different countries in order to gain a better understanding of global trends related to this topic. -

At the end of the discussion chapter, there is a Limitation section that indicates the need to collect more data from various academic centers to understand global trends.

The Conclusions may be somewhat sparse. They should be expanded briefly. In addition, I suggest that paragraphs should be at least 5-6 lines long. In the Conclusions, there are paragraphs with only two lines. - As suggested, the conclusions of the thesis have been extended.

Finally, regarding the Title of the manuscript, I consider it to be too long, and an effort should be made to reduce its length to 12-16 words. This will improve its positioning. Also, I ask for a brief discussion among the manuscript’s authors on the convenience of keeping the word Covid-19 in the title. As I have commented in the Introduction section, please join the comments made in the Introduction section. After all, Covid-19 may no longer be a topic of academic interest, but similar situations may arise in the future.  - Shortened the title and dropped COVID-19 in the title (added to keywords)

Finally, please consider whether the citation form of the references is correct. You are using brackets instead of square brackets. Check this term with the journal template. This would be the main reason for considering the manuscript as "Major revisions". - Round brackets changed to square brackets

Reviewer 2 Report

TOPIC:

Assessment of the Level of Physical Activity and Mental Well-Being in Students after a Year of the Study Conducted in a Mixed Mode under the Conditions of Restrictions Caused by COVID-19

Thank you for inviting me to review this article. My comments are as follows:

TITLE

The title is satisfactory and represents the whole article

ABSTRACT

1.      The abstract should consist of several elements, namely the title statement, research problem, research objective, methodology and research findings. The author, however, started the abstract with the objective of the study without touching on the title statement or problem statement.

2.      Line 11: “Aim of the study: Assessment of the level of physical activity…” The way abstracts are written needs to be improved. It is better to express and organize ideas without using colons (:)

INTRODUCTION

The introduction is satisfactory

MATERIALS AND METHODS

1.      Sample is satisfactory

2.      It is better if the author combined the semi-structured interview with the questionnaire in this study

RESULTS

1.      The discussion and interpretation of the results are fascinating.

DISCUSSION

1.       Line 228 and 242: “In my research..” The statement needs to be amended by using a more appropriate word (i.e. In this research).

2.       Line 255 and 256: “According to WHO, you should also do exercises to strengthen muscles with moderate or greater intensity involve all main muscle groups for 2 or more days a week”. The word "you" should be replaced with another word that fits the context of the sentence.

-Overall, the discussion in this study is insightful, but the presentation of the ideas and sentences is flawed, and needs to be improved.

Author Response

Thank you for your time and attention, we found all the suggestions valuable and we have done our best to improve our manuscript. Below, you will find our answer and amendments changes.

TOPIC:

Assessment of the Level of Physical Activity and Mental Well-Being in Students after a Year of the Study Conducted in a Mixed Mode under the Conditions of Restrictions Caused by COVID-19

Thank you for inviting me to review this article. My comments are as follows:

TITLE

The title is satisfactory and represents the whole article

ABSTRACT

  1. The abstract should consist of several elements, namely the title statement, research problem, research objective, methodology and research findings. The author, however, started the abstract with the objective of the study without touching on the title statement or problem statement. - Introductory sentences to the abstract have been added

  1. Line 11: “Aim of the study: Assessment of the level of physical activity…” The way abstracts are written needs to be improved. It is better to express and organize ideas without using colons (:) - The colon has been omitted.

INTRODUCTION

The introduction is satisfactory

MATERIALS AND METHODS

  1. Sample is satisfactory

  1. It is better if the author combined the semi-structured interview with the questionnaire in this study - replaced

RESULTS

  1. The discussion and interpretation of the results are fascinating.

DISCUSSION

  1. Line 228 and 242: “In my research..” The statement needs to be amended by using a more appropriate word (i.e. In this research). – wording has been changed

  1. Line 255 and 256: “According to WHO, you should also do exercises to strengthen muscles with moderate or greater intensity involve all main muscle groups for 2 or more days a week”. The word "you" should be replaced with another word that fits the context of the sentence. - – wording has been changed

-Overall, the discussion in this study is insightful, but the presentation of the ideas and sentences is flawed, and needs to be improved. - The discussion has been re-read, errors found have been corrected

Reviewer 3 Report

This article addresses the links between covid-19 and students' well-being under restrictions. They specifically focused on the hypothesis that covid -19 increases the risk of depression in college students. This is a small study that may contribute to public health research. Although they used standard tools to measure indicators of well-being and mental health, this paper has a flaw in the small population.

Concerns:

The small population is used for data analysis. A similar paper with a large population sampling had shown that restriction can cause depression and other mental health problems.

The study design is not clear and lacks how the questionnaires were acquired. How was the participant evaluated? How do they exclude bias? How many people submitted an incomplete questionnaire?

How was the measured activity? by questionnaire?

Other measures to make more evident the mental health disequilibrium:

Socio- economic-demographic characteristics, body health status, comprehensive mental health evaluation, energy levels during the normal routine, and physical habits that change during restriction.  

Any covariates?

Data analysis? Mediation analysis, intra correlations, class correlations.

How do you validate data analysis?

It doesn’t have a clear hypothesis.

The title indicates mental well-being, but most of the paper focuses on depression. Maybe change accordingly the paper title.

What are the strength and limitations?

The conclusion doesn’t reflect the whole paper.

Author Response

Thank you for your time and attention, we found all the suggestions valuable and we have done our best to improve our manuscript. Below, you will find our answer and amendments changes.

This article addresses the links between covid-19 and students' well-being under restrictions. They specifically focused on the hypothesis that covid -19 increases the risk of depression in college students. This is a small study that may contribute to public health research. Although they used standard tools to measure indicators of well-being and mental health, this paper has a flaw in the small population. – Limitations has been added

Concerns:

The small population is used for data analysis. A similar paper with a large population sampling had shown that restriction can cause depression and other mental health problems. -  Taking into account the research using questionnaires, the sample size is actually not large. This notice is included in the Limitation section. The presented research also showed that a significant number of students had a depressed mood (30% of group 1 and 45% of group 2) (line 162-167).

The study design is not clear and lacks how the questionnaires were acquired. How was the participant evaluated? How do they exclude bias? How many people submitted an incomplete questionnaire? - Recruitment for the research was carried out through student social media. Information about recruitment for research and how to participate was also provided during student classes. The qualification for the study was the status of a full-time student and the declared absence of chronic physical and mental illnesses in the last year. The respondents filled in the online version of the questionnaire (google survey), and the results in the form of answers were automatically transferred by the system to a spreadsheet. All started surveys were complete. Line 90-95

How was the measured activity? by questionnaire? - Yes., Physical activity was assessed qualitative using the Global Physical Activity Questionnaire (GPAQ) recommended for this type of analysis by WHO

Other measures to make more evident the mental health disequilibrium:-

Socio- economic-demographic characteristics, body health status, comprehensive mental health evaluation, energy levels during the normal routine, and physical habits that change during restriction.   - In fact, the research did not take into account socio-economic characteristics and a comprehensive assessment of mental health. The work was supplemented with a limitation section in which this information was included.

The participants were theoretically healthy physically and mentally. This is due to the inclusion and exclusion criteria. The subjects in the first section of the questionnaire answered 2 questions about physical and mental health. 100% of respondents to both questions answered "no disease processes in the period of 1 year preceding the survey.

The normal routine (time of physical activity, time of sitting and sleeping) was analyzed by the Global Physical Activity Questionnaire (GPAQ).

Any covariates? Data analysis? Mediation analysis, intra correlations, class correlations. - In the study, it was decided to use logistic regression in order to determine which variables increase the probability of depression. checked min. impact of gender, physical activity, sleep duration, sitting time, COVID-19 incidence. In addition, the odds ratio was calculated. In the process of data analysis, a multi-factor regression model was also performed. It showed similar dependencies as the logistic model, so it was not included in the paper.

However, correlations were not analysed. Due to the ordinal nature of the variables or the lack of normal distribution of most quantitative variables, data variability was checked using the interquarter range (IQR). For the same reason, intraclass correlation and analysis of variance, which require the use of the arithmetic mean, were not used. Depending on the comparison, Cohen's d or Cramer's coefficient was used as the measure of effect

How do you validate data analysis? - In order to verify the data analysis, the effect size values calculated depending on the tested test using Cohen's d or Cramer's coefficient were added

It doesn’t have a clear hypothesis. - A hypothesis was added before the aim of the work

The title indicates mental well-being, but most of the paper focuses on depression. Maybe change accordingly the paper title. – title has been shortened and rewritten

What are the strength and limitations? Section Limitations has been added

The conclusion doesn’t reflect the whole paper. - the conclusions section has been redrafted

Round 2

Reviewer 1 Report

Dear authors.

Thank you for incorporating the proposed revisions. I believe that the manuscript meets the conditions for publication.

Author Response

Answer to Reviewer 1 round second

One more time thank you for your time and valuable comments.

 Kind regards

authors

Reviewer 3 Report

In their response to my previous comments, the authors clarified most of the questions. New statistical analysis was added to improve data presentation and analysis. However, the authors forget to interpret the new statistical analysis in this draft.

  1. Two different analyses were performed (Cohen’s d and Cramer’s V), and data was added to the graphics, but it lacks interpretation of that data or new analysis under the results section.
  2. Although, the Chi2 shows statistical, the new data shows low effects.
  3. Some data shows Cramer’s V/ Cohens’d, it needs to be addressed in the statistical section. Not all reader has statistical literacy.
  4. Nothing is mentioned about the new analysis in the discussion section.
  5. “No significant relationship was found between the result showing a depressed mood” why? The paper address mood, it needs to be addressed in the conclusion section.

Author Response

 Answer to reviewer  second round 3

Thank you for your time and attention, we found all the suggestions valuable and we have done our best to improve our manuscript. Below, you will find our answer and amendments changes.

In their response to my previous comments, the authors clarified most of the questions. New statistical analysis was added to improve data presentation and analysis. However, the authors forget to interpret the new statistical analysis in this draft.

  1. Two different analyses were performed (Cohen’s d and Cramer’s V), and data was added to the graphics, but it lacks interpretation of that data or new analysis under the results section. - Added description in results section line 226-236

  1. Although, the Chi2 shows statistical, the new data shows low effects.- Added description in results section line 226-236
  2. Some data shows Cramer’s V/ Cohens’d, it needs to be addressed in the statistical section. Not all reader has statistical literacy. - The description of statistical methods has been supplemented

  1. Nothing is mentioned about the new analysis in the discussion section. - Describing the results of own research, reference was also made to the effect size.
  2. “No significant relationship was found between the result showing a depressed mood” why? The paper address mood, it needs to be addressed in the conclusion section.  - The conclusions section has been supplemented for amendment information.

Round 3

Reviewer 3 Report

Table 1 shows a p-value but no p-value was used to interpret the table.

Add the actual Chi2 test value and next to it the interpreted P-value or add it in the statistical analysis section.

Need a little text editing. Line 155

Line 16… student mood

Line 25 was not corrected.  It should be mental health or mood.  

Author Response

Thank you for your time and attention. Below, you will find amendment changes.

Table 1 shows a p-value but no p-value was used to interpret the table - A reference to p-values has been added to the text

Add the actual Chi2 test value and next to it the interpreted P-value or add it in the statistical analysis section. - The Ch2 test value has been added to the tables

Need a little text editing. Line 155 - Sentence corrected

Line 16… student mood - corrected

Line 25 was not corrected.  It should be mental health or mood. - changed
